# Graph based Consistency Learning for Contrastive Multi-View Clustering

## ABSTRACT

Multi-View Clustering (MVC) aims to mine complementary information across different views to partition multi-view data more effectively and has attracted considerable interest. However, existing deep multi-view clustering methods frequently neglect the exploration of structural information within individual view and lack the learning of structural consistency among views, which results in limitations in the clustering performance. In this paper, we introduce a novel multi-view clustering framework based on graph consistency learning to address this issue. Specifically, we design intra-view graph contrastive learning to uncover structural information within each view and achieve structural conscistency objectives through cross-view graph consistency learning. Additionally, to address the conflict between different learning objectives when trained in the same space, we introduce two new feature spaces, one for cluster-levcel contrastive learning and the other for instance-level contrastive learning. Subsequently, to make the most of discriminative information from all views, we concatenate high-level features from all views to form global features and employ self-supervision to promote clustering consistency across different views. Experimental results on several challenging datasets demonstrate the outstanding performance of our proposed method.

## CCS CONCEPTS

• **Multimodal Fusion and Embeddings → Multi-View Clustering**.

## KEYWORDS

Deep Multi-View Clustering, Contrastive Clustering, Graph Learning.

**ACM Reference Format:**
Anonymous Author(s). 2024. Graph based Consistency Learning for Contrastive Multi-View Clustering. In *Proceedings of (MM '24)*. ACM, New York, NY, USA, 9 pages. https://doi.org/XXXXXXX.XXXXXXX

## 1 INTRODUCTION

Clustering analysis is a fundamental unsupervised learning task in machine learning, widely applied in fields such as computer vision and data mining. Its primary objective is to partition data items with similar features into the same group in the absence of label information. In the real world, multi-view data is prevalent

in many practical applications, originating from different information sources or diverse feature extraction methods. For instance, a news story can be reported from various angles by different media outlets, and a bouquet of flowers can be described from multiple dimensions, including color, fragrance and species. Because different views can offer distinct data perspectives, relyin g solely on information from a single view is often insufficient. In recent years, Multi-View Clustering (MVC) has garnered significant attention as a vital research domain in unsupervised learning. The primary goal of MVC is to extract complementary and consistent information from multiple views to enhance clustering performance.

In the early stages, MVC primarily employed conventional machine learning methods for clustering analysis. These approaches can be broadly categorized into graph-based learning method[19, 20, 30], multi-kernel method[25, 26, 41], subspace clustering mothed[8, 10, 35–37] and non-negative matrix factorization[2, 14, 15] method. For graph-based learning mothod, multiple graphs are used to represent relationships among data from different views, enabling the exploration of structural consistency across multiple views. Multi-kernel method utilizes kernel functions to unveil underlying clustering patterns at the view level. Subspace clustering mothed predominantly aims to find a shared subspace for representation learning. Non-negative matrix factorization method, on the other hand, focuses on dimensionality reduction and factorization of the feature matrix to achieve more effective feature representations. Nevertheless, in the real-world scenarios, many traditional MVC methods suffer from weak representational capability and high computational complexity, ultimately limiting their applications.

In recent years, due to the powerful nonlinear fitting capabilities of deep neural networks, many researches focus on leveraging deep models for MVC to overcome the limitations of traditional machine learning. Specifically, MVC methods based on deep representations implement deep neural networks as nonlinear parametric mapping functions, effectively exploring the nonlinear characteristics embedding in the original data space. In deep multi-view clustering, one of the most commonly used models is the autoencoder, typically composed of a symmetric Multi-Layer Perceptron (MLP), which effectively preserves feature information in the original data space through reconstruction loss. Recent studies integrate multi-view feature learning and clustering assignment into a unified framework, allowing clustering results to participate in network training to improve the quality of feature learning. This results in an end-to-end multi-view clustering framework that achieves excellent clustering performance[21, 32, 40].

Despite significant advancements in deep MVC methods in recent years, they still face limitations and challenges: (1) The neglect of exploring view information from a structural perspective, while focusing solely on instance-level or cluster-level consistency, can lead to the disruption of structure information and the loss of sample association information. It may increase the difficulty of clustering.

(2) Some multi-view clustering methods attempt to simultaneously achieve the consistency of different objectives within a single feature space, usually by merely stacking loss functions. It may result in less discriminative feature representations, making it challenging to capture complex data structures effectively. (3) Many approaches involve a two-stage multi-view clustering process, where traditional clustering algorithms (e.g., K-means[16] or spectral clustering[17]) are applied after multi-view feature learning to obtain the final clustering results. It makes that, clustering cannot guide the feature learning process, limiting overall performance improvements. Hence, addressing these challenges in deep multi-view clustering is crucial for improving clustering performance and the effectiveness of feature learning.

In this paper, to address the challenges mentioned above, we propose a novel deep multi-view clustering framework termed Graph based Consistency learning for Contrastive Multi-View Clustering (GC-CMVC). The framework consists of three key modules: graph learning, hierarchical contrastive learning, and self-supervised clustering. The graph learning module aims to dig deeper into the local structure information within each view through contrastive learning and capture structurally consistent feature representations among views more effectively through graph consistency learning. The hierarchical contrastive learning module incorporates two MLP shared by all the views to obtain high-level features and cluster distributions. It performs instance-level and cluster-level consistency learning in their respective feature spaces. In the self-supervised clustering module, high-level features are concatenated to construct global features and generate pseudo-labels for self-supervised learning. These modules are seamlessly integrated and collaborated to enhance multi-view clustering. Our contributions are listed as follows:

(1) We develop an end-to-end deep multi-view clustering framework, incorporating graph learning to better explore local geometric information within each view. By performing consistency learning on relationship graphs across different views, we successfully capture the consistent structure information within multi-view data.

(2) We introduce distinct objectives in different feature spaces, performing multi-level contrastive learning at both the cluster and instance levels. At the same time, we construct global features to guide the clustering distribution of each view through self-supervised learning.

(3) Extensive experiments on various datasets demonstrate the remarkable effectiveness and superior performance of our method in MVC tasks.

## 2 RELATED WORK

### 2.1 Deep Multi-View Clustering

In the realm of multi-view clustering, the advent of deep multi-view clustering has drawn considerable attention, benefiting from the nonlinear modeling capabilities of neural networks. These methods primarily leverage deep learning architectures for feature extraction, and are successfully applied in various real-world scenarios.

Deep MVC methods can be divided into three distinct categories: (1) Subspace-based Methods[24, 42]. This category of techniques encodes each view of the original data using autoencoders and subsequently combines the encoded features through self-expression to obtain comprehensive representations. Notably, Zhu et al.[42] employed Diversity Networks (Dnet) and Universality Networks (Unet) to learn view-specific self-expressive matrices and a common self-expressive matrix for all views. To preserve view-specific structural information, Zheng et al.[38] introduced first-order and second-order graph mining, capturing local and global graph information to guide subspace representation learning. (2) Generative Model-based Methods. This category of methods aims to learn underlying data patterns and features to generate new samples akin to the original data. Li et al.[12] used autoencoders as generators in a Generative Adversarial Network (GAN), incorporating a fully connected network as a discriminator for post-decoding. To enhance feature representation ability, Zhou et al.[40] introduced a novel Cauchy-Schwarz divergence-based clustering loss that encourages both cluster separation and intra-cluster compactness. (3) Contrastive Learning-based Methods. This category of approach excels at learning shared representations across different views by maximizing the similarity of positive samples within views and minimizing the similarity of negative samples across views. Chen et al.[3] introduced a methodology that amalgamates contrastive loss and clustering loss from three modalities, offering a comprehensive constraint on feature learning.

### 2.2 Contrastive Clustering

Contrastive learning has exceptional representation learning capabilities and is widely applied in various domains[4, 18, 27]. Its core concept revolves around constructing pairs of positive and negative samples, aiming to maximize the similarity of positive pairs and minimize that of negative pairs in the feature space to learn embedding representations. Many works employ contrastive learning strategies to data clustering and obtains excellent performance. Li et al.[11] proposed an end-to-end online image clustering method termed Contrastive Clustering (CC), which introduces two novel feature spaces for instance-level contrastive learning and cluster-level contrastive learning, and generate more accurate image embedding representations for clustering. Zhong et al.[39] presented a novel graph contrastive clustering method termed Graph Contrastive Clustering (GCC). GCC addresses the limitations of existing contrastive learning methods by considering category information and clustering objectives, yielding representations better suited for clustering tasks. To address the issue of false positives, Yin et al.[34] leveraged an effective data augmentation method ContrastiveCrop and constructed positive sample pairs based on nearest-neighbor mining to acquire more semantic information. Contrastive clustering is also generalized to multi-view data, and lots of contrastive multi-view clustering methods are proposed[13, 21, 33]. To extend contrastive learning to the multi-view domain, Trosten et al.[21] employed instance-level contrastive learning loss to further enhance the clustering performance of the model. Lin et al.[13] utilized contrastive learning based on cross-view mutual information to obtain informative and consistent representations. Xu et al.[33] proposed multi-view instance-level contrastive learning and clustering-level contrastive learning, seamlessly integrating them into a unified framework.

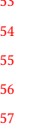

**Figure 1: The framework of GC-CMVC. GC-CMVC consists of three main modules: graph learning (intra-view graph contrastive learning and cross-view graph consistency learning), hierarchical contrastive learning (cluster-level contrastive learning and instance-level contrastive learning), and self-supervised clustering . These modules work together to explore intra-view consistency and complementary information, which aims to achieve consistency for clustering results across different feature spaces.**

## 3 PROPOSED METHOD

**Problem Description:** Given a clustering problem with a set of $N$ samples composed of $V$ views, the goal is to partition $D = \{X^1, \ldots, X^v, \ldots, X^V\}$ into $C$ clusters, where $X^v = \{x_1^v, \ldots, x_i^v, \ldots, x_N^v\} \in \mathcal{R}^{d_v \times N}$ represents $N$ samples with dimension $d_v$ from the $v$-th view. Our model is illustrated in Figure 1, and we will now explain each module one by one.

### 3.1 Specific View Reconstruction

Considering that data from different views typically have distinct dimensions and may contain redundancies and random noise, we employ a set of view-specific deep autoencoders. These autoencoders are designed to ensure that the learned embedding features contain the characteristic information of the original data space. Specifically, for the $v$-th view, we denote $f_{\theta^v}^v$ and $g_{\phi^v}^v$ as the encoder and decoder, with $\theta^v$ and $\phi^v$ representing the trainable parameters of autoencoders. The notation $z_i^v = f_{\theta^v}^v(x_i^v)$ represents the low-dimensional and more aggregated embedding features of $x_i^v$ obtained by the encoder network. To train the autoencoder network , we define the multi-view reconstruction loss as follows:

$$L_{RC} = \sum_{v=1}^{V} \sum_{i=1}^{N} \| x_i^v - g_{\phi^v}^v(f_{\theta^v}^v(x_i^v)) \|_2^2 . \quad (1)$$

Through the autoencoder, we obtain low-dimensional embedding features $Z^v = \{z_1^v, \ldots, z_i^v, \ldots, z_N^v\}$ from the high-dimensional and information-dispersed data $X^v$.

### 3.2 Graph Learning

**Graph Contrastive Learning:** In order to fully explore the local structure within each view and retain the structural information of the original data in the embedding feature space, we construct a relationship graph $G_i^v = \{z_{i1}^v, \ldots, z_{ik}^v, \ldots, z_{iK}^v\}$ containing $K$ nearest neighbors of $z_i^v$. In other words, we consider that, within the $v$-th view, $z_i^v$ is strongly associated with the samples in $G_i^v$. During the network training process, their features should be as similar to $z_i^v$ as possible. Simultaneously, for samples not in $G_i^v$, they should be kept as dissimilar to $z_i^v$ as possible. In the context of contrastive learning, if a sample is presented in the relationship graph of another sample, they form a positive pair in the graph structure; otherwise, they constitute a negative pair. We measure the similarity between different samples using cosine similarity:

$$s(z_i^v, z_j^v) = \frac{(z_i^v)^T (z_j^v)}{\| z_i^v \| \| z_j^v \|}. \quad (2)$$

Therefore, the intra-view graph contrastive learning loss for $z_i^v$ is defined as follows:

$$L_{GC_i}^v = -\frac{1}{K} \sum_{k=1}^{K} \log \frac{e^{s(z_i^v, z_{ik}^v)/\tau_G}}{\sum_{j=1}^{N} e^{s(z_i^v, z_j^v)/\tau_G} - e^{1/\tau_G}}, \quad (3)$$

where $\tau_G$ denotes the temperature parameter, $z_{ik}^v$ denotes $k$-th neighbor sample of $z_i^v$, $s(z_i^v, z_j^v)$ is the cosine distance used to measure the similarity between two embedding features. Through this loss, $z_i^v$ is encouraged to be more close to samples within the relationship graph and farther from other samples. With all the training samples, the overall loss for intra-view graph contrastive learning is defined as follows:

$$L_{GC} = \frac{1}{NV} \sum_{v=1}^{V} \sum_{i=1}^{N} L_{GC_i}^v. \quad (4)$$

**Graph Consistency Learning:** While intra-view graph contrastive learning fully explores local structure information within each view, MVC requires a strong emphasis on the consistency of information across different views. Therefore, it is insufficient to retain only local structural information. To address this issue, we introduce a method for learning structural consistency among different views[28]. Specifically, we constrain the relationship graph of the same sample in different views to be as similar as possible, encouraging the structural information of different views to be more consistent. The graph consistency learning for $z_i^v$ can be expressed as follows:

$$Lcc_i^v = \frac{1}{K} \sum_{j \neq v}^{V} \sum_{k=1}^{K} \| z_{ik}^v - z_{ik}^j \|_2^2, \quad (5)$$

$z_{ik}^v$ and $z_{ik}^j$ belong to $G_i^v$ and $G_i^j$, representing neighbors of $z_i^v$ in the $v$-th and $j$-th views, respectively. The overall cross-view consistency learning is defined as follows:

$$L_{CC} = \frac{1}{NV} \sum_{i=1}^{N} \sum_{v=1}^{V} Lcc_i^v. \quad (6)$$

In summary, our graph learning loss $L_G$ consists of $L_{CC}$ and $L_{GC}$, and is defined as:

$$L_G = L_{CC} + L_{GC}. \quad (7)$$

## 3.3 Hierarchical Contrastive Learning

To address the conflict between different learning objectives when trained in the same feature space, we introduce a hierarchical contrastive learning to achieve both instance-level and cluster-level consistency objectives.

**Instance-Level Contrastive Learning:** In multi-view scenarios, there are naturally positive and negative sample pairs where the feature representations of each sample across different views are considered as positive pairs, and other combinations are regarded as negative pairs. Based on these sample pairs, we utilize a three-layer MLP network on top of the embedding features to obtain high-level features $\{H^v\}_{v=1}^V$. The dimensionality of high-level features is usually smaller than that of the embedded features. We introduce a contrastive learning loss for instance-level consistency to learn the commonality among these high-level features, which helps capture

the instance-level consistency information of multi-view data more effectively, the instance-level contrastive loss is defined as:

$$L_S^{ab} = -\frac{1}{N} \sum_{i=1}^{N} \log \frac{e^{s(h_i^a, h_i^b)/\tau_S}}{\sum_{j=1}^{N} \sum_{v=a,b} e^{s(h_i^a, h_j^b)/\tau_S} - e^{1/\tau_S}}, \quad (8)$$

where $\tau_S$ denotes the temperature parameter. We extend the instance-level contrastive learning loss to the multi-view scenario, and the loss is defined as follows:

$$L_S = \frac{1}{V} \sum_{a=1}^{V} \sum_{b \neq a}^{V} L_S^{ab}. \quad (9)$$

**Cluster-Level Contrastive Learning:** To achieve cluster-level consistency, similar to instance-level contrastive learning, we also stack a three-layer MLP network on the embedding features. The difference lies in the final layer, where we use the softmax function to obtain the probability distribution for cluster assignment, i.e., $\{Q_{\cdot j}^v = \{q_{ij}^v\}_{i=1}^N\}_{v=1}^V$, where $q_{ij}^v$ represents the probability that the $i$-th sample in the $v$-th view belongs to the $j$-th cluster. The objective of cluster consistency is to ensure that the distributions of samples from the same cluster are as similar as possible, while the distributions between different clusters are as dissimilar as possible. The contrastive learning loss function for $Q^{(a)}$ and $Q^{(b)}$ is formulated as follows:

$$L_Q^{ab} = -\frac{1}{C} \sum_{j=1}^{C} \log \frac{e^{s(Q_{\cdot j}^a, Q_{\cdot j}^b)/\tau_L}}{\sum_{c=1}^{C} \sum_{v=a,b} e^{s(Q_{\cdot j}^a, Q_{\cdot c}^b)/\tau_L} - e^{1/\tau_L}}, \quad (10)$$

where $\tau_L$ denotes the temperature parameter. Similar to the instance-level contrastive learning, the multi-view clustering-level contrastive learning loss is defined as follows:

$$L_Q = \frac{1}{V} \sum_{a=1}^{V} \sum_{b \neq a}^{V} L_Q^{ab} + \sum_{v=1}^{V} \sum_{j=1}^{C} s_j^v \log s_j^v, \quad (11)$$

where $s_j^v = \frac{1}{N} \sum_{i=1}^{N} q_{ij}^v$. The latter term is a regularization component aimed at preventing all samples from being assigned to a single cluster[23].

To sum up, our hierarchical contrastive learning loss consists of $L_S$ and $L_Q$, and is defined as:

$$L_{HCL} = L_S + L_Q. \quad (12)$$

## 3.4 Self-Supervised Clustering

In multi-view clustering, the discriminative power of samples is varied between different views, with some views being more effective at distinguishing between different samples. When we concatenate the features from each view, those views with higher feature discriminability often play a more important role during the sample discrimination process. Based on this principle, to make the most of discriminative information from all views, we concatenate all the high-level features obtained from instance-level contrastive learning to generate a global feature representation $h_i$ as:

$$h_i = [h_i^1; h_i^2; ...; h_i^V] \in \mathbb{R}^{\sum_{v=1}^{V} d_v}. \quad (13)$$

Next, we apply K-means to the global features to compute the cluster centers, denoted as $\mu_j$, where j represents the $j$-th cluster. And then we use the t-distribution[22] to measure the similarity

between the global feature $h_i$ and the cluster center $\mu_j$, which is defined as:

$$t_{ij} = \frac{(1 + ||h_i - \mu_j||^2)^{-1}}{\sum_{j=1}^{C}(1 + ||h_i - \mu_j||^2)^{-1}}. \tag{14}$$

Normally, components with high probabilities in the soft assignment represent high confidence. To enhance the discriminability of the soft assignment, we employ the following method to sharpen them to obtain a consistent target distribution $p_{ij}$ as:

$$p_{ij} = \frac{t_{ij}^2/\sum_{i=1}^{N} t_{ij}}{\sum_{j=1}^{C}(t_{ij}^2/\sum_{i=1}^{N} t_{ij})}. \tag{15}$$

To ensure the entire model can achieve clustering consistency across various feature spaces, we perform self-supervised training with consistent probability distribution and special probability distribution of each view. Specifically, for the $v$-th view, we introduce a clustering loss to measure the Kullback-Leibler divergence between the unified target distribution $p_{ij}$ and the clustering distribution $q_{ij}^v$ of each view, which is defined as:

$$L_C = \sum_{v=1}^{V}\sum_{i=1}^{N}\sum_{j=1}^{C} p_{ij} \log \frac{p_{ij}}{q_{ij}^v}. \tag{16}$$

Finally, we obtain the clustering labels $\{y_i\}_{i=1}^{N}$ based on $p_{ij}$ as:

$$y_i = \underset{j}{\arg\max}\, p_{ij}. \tag{17}$$

## 3.5 Optimization

We integrate the aforementioned four modules within an end-to-end deep learning framework. The view reconstruction $L_{RC}$ loss is constructed on embedded feature $\{Z^v\}_{v=1}^{V}$. The graph learning loss $L_G$ is constructed based on relationship graph $\{G_i^v\}_{v=1}^{V}$ to preserve structural information. Hierarchical contrastive learning loss $L_{HCL}$ is employed to respectively obtain high-level features $\{H^v\}_{v=1}^{V}$ and clustering distribution $\{Q_{\cdot j}^v\}_{v=1}^{V}$. Self-supervised clustering loss $L_C$ is imposed on $\{H^v\}_{v=1}^{V}$ and $\{Q_{\cdot j}^v\}_{v=1}^{V}$ to learn consistent cluster assignments. With these losses mentioned above, the overall loss of our model is defined as:

$$
\begin{aligned}
L &= L_{RC} + \lambda_1 L_{HCL} + \lambda_2 L_G + L_C \\
&= L_{RC}\left(\{X^v, \hat{X}^v\}_{v=1}^{V}; \{\theta_v, \phi_v\}_{v=1}^{V}\right) \\
&\quad + \lambda_1 L_{HCL}\left(\{H^v, Q^v\}_{v=1}^{V}; \{\psi, \epsilon\}\right) + \lambda_2 L_G + L_C
\end{aligned}
\tag{18}
$$

where $\lambda_1$ and $\lambda_2$ are hyperparameters used to balance the losses between hierarchical contrastive learning and graph learning. The optimization process of GC-CMVC is summarized in Algorithm 1.

## 4 EXPERIMENTS

### 4.1 Datasets

As shown in Table 1, the experiments are conducted on a variety of representative datasets, including RGB-D[40], Columbia Consumer Video (CCV)[7], Fashion[31], Reuters[1] and Caltech[5]. The introduction of these datasets is given as follows:

RGB-D[40] consists of 1,449 images from 13 indoor scenes, with each image accompanied by descriptive paragraphs providing rich multi-view information.

---

**Algorithm 1:** Algorithm of GC-CMVC

**Input** : Multi-view dataset $\{X^v\}_{v=1}^{V}$; Number of clusters C; Temperature parameters $\tau_G$, $\tau_S$ and $\tau_L$; K-Nearest neighbor parameter K; Hyperparameters $\lambda_1$ and $\lambda_2$.

**Output** : Clustering distribution P; Global features $\{h_i\}_{i=1}^{N}$; Clustering label $\{y_i\}_{i=1}^{N}$.

1  Initialize $\{\theta^v, \phi^v\}_{v=1}^{V}$ by minimizing Eq.(1).

2  Concatenate all the high-level features obtained from instance-level contrastive learning to generate global features $\{h_i\}_{i=1}^{N}$.

3  Initialize $\{\mu_j\}_{j=1}^{C}$ by K-means on $\{h_i\}_{i=1}^{N}$.

4  **while** *not converged* **do**

5    Construct a relationship graph $\{G^v\}_{v=1}^{V}$ by K-nearest neighbor on $\{Z^v\}_{v=1}^{V}$.

6    Optimize $\{\{\theta^v, \phi^v\}_{v=1}^{V}, \psi, \epsilon\}$ by minimizing Eq.(18).

7  Output the clustering label $\{y_i\}_{i=1}^{N}$ by Eq.(17).

---

| Dataset | Sample | Class | View | Feature |
|---------|--------|-------|------|---------|
| RGB-D | 1449 | 13 | 2 | 2048/300 |
| CCV | 6773 | 20 | 3 | 5000/5000/4000 |
| Fashion | 10000 | 10 | 3 | 784/784/784 |
| Reuters | 18758 | 6 | 2 | 10/10 |
| Caltech-2V | 1400 | 7 | 2 | 40/254 |
| Caltech-3V | 1400 | 7 | 3 | 40/254/928 |
| Caltech-4V | 1400 | 7 | 4 | 40/254/928/512 |
| Caltech-5V | 1400 | 7 | 5 | 40/254/928/512/1984 |

**Table 1: Specification and partitioning of the selected datasets.**

CCV[7] consists of 9,317 YouTube videos, covering 20 distinct semantic categories.

Fashion[31] is an image dataset related to products, where three different styles are treated as three views of the product.

Reuters[1] is a subset of a text dataset, containing 18,758 samples from six different categories. It provides large-scale data for multi-view text clustering tasks.

Caltech[5] is an RGB image dataset with multiple views. To evaluate the robustness of our method under varying numbers of views, four different versions of Caltech, namely Caltech-2V, Caltech-3V, Caltech-4V and Caltech-5V, are created, which consist of 2, 3, 4 and 5 views respectively[33].

### 4.2 Comparison Methods

The comparison methods can be broadly categorized into two classes. The first class is traditional MVC methods, including MVC-LFA [26], COMIC[19] and IMVTSC-MVI[30]. The second class is advanced deep learning methods, including RMSL[10], CDIMC-net [29], EAMC [40], SiMVC&CoMVC [21] and MFLVC [33]. Their corresponding descriptions are given as follows:

**MVC-LFA** [26]: The mothed aims to maximally align the consensus partition with the weighted base partitions, thereby significantly reducing computational complexity and simplifying the optimization process.

| Datasets | RGB-D | | | CCV | | | Fashion | | | Reuters | | |
|---|---|---|---|---|---|---|---|---|---|---|---|---|
| Metrics | ACC | NMI | PUR | ACC | NMI | PUR | ACC | NMI | PUR | ACC | NMI | PUR |
| RMSL[10] | 31.4 | 24.5 | 32.7 | 21.5 | 15.7 | 24.3 | 40.8 | 40.5 | 42.1 | 33.6 | 16.0 | 31.1 |
| MVC-LFA [26] | 37.9 | 39.8 | 39.7 | 23.2 | 19.5 | 26.1 | 79.1 | 75.9 | 79.4 | 41.9 | 20.3 | 42.0 |
| COMIC[19] | 31.2 | 28.6 | 32.0 | 15.7 | 8.1 | 15.7 | 57.8 | 64.2 | 60.8 | 33.8 | 14.9 | 32.3 |
| IMVTSC-MVI[30] | 35.5 | 31.2 | 36.4 | 11.7 | 6.0 | 15.8 | 63.2 | 64.8 | 63.5 | 40.9 | 21.4 | 41.0 |
| CDIMC-net [29] | 39.2 | 35.4 | 38.7 | 20.1 | 17.1 | 21.8 | 77.6 | 80.9 | 78.9 | 39.7 | 20.1 | 41.2 |
| EAMC [40] | 32.3 | 20.7 | 32.3 | 26.3 | 26.7 | 27.4 | 61.4 | 60.8 | 63.8 | 41.3 | 27.8 | 42.7 |
| SiMVC[21] | 39.6 | 35.6 | 38.7 | 15.1 | 12.5 | 21.6 | 82.5 | 83.9 | 82.5 | 45.5 | 26.4 | 45.5 |
| CoMVC[21] | 41.3 | **40.5** | 41.0 | 29.6 | 28.6 | 29.7 | 85.7 | 86.4 | 86.3 | 48.4 | 23.6 | 48.0 |
| MFLVC[33] | 37.6 | 24.7 | 43.8 | **31.2** | **31.6** | 33.9 | **99.2** | 98.0 | **99.2** | 44.9 | 23.8 | 49.9 |
| OURS | **46.4** | 30.2 | **49.6** | 30.4 | 30.1 | **34.0** | 99.0 | **98.5** | **99.5** | **54.6** | **31.5** | **62.0** |

**Table 2: Clustering results on four multi-view datasets. Bold denotes the best results and underline denotes the second-best.**

| Datasets | Caltech-2V | | | Caltech-3V | | | Caltech-4V | | | Caltech-5V | | |
|---|---|---|---|---|---|---|---|---|---|---|---|---|
| Metrics | ACC | NMI | PUR | ACC | NMI | PUR | ACC | NMI | PUR | ACC | NMI | PUR |
| RMSL[10] | 52.5 | 47.4 | 54.0 | 55.4 | 48.0 | 55.4 | 59.6 | 55.1 | 60.8 | 35.4 | 34.0 | 39.1 |
| MVC-LFA [26] | 46.2 | 34.8 | 49.6 | 55.1 | 42.3 | 57.8 | 60.9 | 52.2 | 63.6 | 74.1 | 60.1 | 74.7 |
| COMIC[19] | 42.2 | 44.6 | 53.5 | 44.7 | 49.1 | 57.5 | 63.7 | 60.9 | 76.4 | 53.2 | 54.9 | 60.4 |
| IMVTSC-MVI[30] | 49.0 | 39.8 | 54.0 | 55.8 | 44.5 | 57.6 | 11.7 | 6.0 | 15.8 | 76.1 | 0.69.1 | 78.5 |
| CDIMC-net [29] | 51.5 | 48.0 | 56.4 | 52.8 | 48.3 | 56.5 | 56.0 | 56.4 | 61.7 | 72.7 | 69.2 | 74.2 |
| EAMC [40] | 41.9 | 25.6 | 42.7 | 38.9 | 21.4 | 39.8 | 35.6 | 20.5 | 37.0 | 31.8 | 17.3 | 34.2 |
| SiMVC[21] | 50.8 | 47.1 | 57.7 | 56.9 | 49.5 | 59.1 | 61.9 | 53.6 | 63.0 | 71.9 | 67.7 | 72.9 |
| CoMVC[21] | 46.6 | 42.6 | 52.7 | 54.1 | 50.4 | 58.4 | 56.8 | 56.9 | 64.6 | 70.0 | 68.7 | 74.6 |
| MFLVC[33] | 60.6 | 52.8 | 61.6 | 63.1 | 56.6 | 63.9 | 73.3 | 65.2 | 73.4 | 80.4 | 70.3 | 80.4 |
| OURS | **64.2** | **53.5** | **64.1** | **68.9** | **60.4** | **70.6** | **76.4** | **72.8** | **78.4** | **84.9** | **76.4** | **84.9** |

**Table 3: Clustering results on Caltech dataset with different views. Bold denotes the best results and underline denotes the second-best.**

**COMIC**[19]: It projects raw data into a unified space, where the projection emphasizes both geometric consistency and cluster assignment consistency.

**IMVTSC-MVI**[30]: A multi-view clustering approach for the challenging problem of multi-view clustering with missing views, recovers missing views and leverage the full information from both recovered and available views for clustering.

**RMSL**[10]: A deep multi-view clustering approach handles high-dimensional data while simultaneously exploring the consistency and complementarity among different views.

**CDIMC-net** [29]: An incomplete multi-view clustering network integrates view-specific deep encoders and graph embedding strategies into its framework, capturing the high-level features and local structure of each view.

**EAMC**[40]: An end-to-end adversarial-attention network for multi-modal clustering, uses adversarial learning and attention mechanisms to align latent feature distributions and quantify the importance of modalities.

**SiMVC&CoMVC** [21]: It employs a weighting strategy to blend representations for the final data clustering process. This strategy involves feature-level contrastive learning of all views.

**MFLVC** [33]: It is a deep contrastive multi-view clustering method, which learns feature representations at different levels of multi-view data within an end-to-end network, and utilizes contrastive learning to achieve feature-level and cluster-level objective consistency.

### 4.3 Implementation Details

To ensure a fair comparison, we set the parameters of each MVC method according to the settings of the original papers. And We evaluate MVC methods using three clustering validity metrics: Accuracy (ACC), Normalized Mutual Information (NMI) and Purity (PUR). These metrics offer a comprehensive measure of clustering quality, with higher values indicating better clustering performance. For our method, we train the proposed network using the PyTorch platform[19]. Adam optimizer[9] is employed to optimize our model, and the initial learning rate is set as 0.0003. For all datasets, we set the batch size as 256 and fix the number of training epochs at 100. Additionally, in the graph contrastive learning process, we compute K-nearest neighbor graphs, with $K$ set as 10. Our model has a simple network architecture with low computational resource requirements. The experiments are conducted on PC with GeForce RTX 3090 GPU, Intel i9-12900F CPU, 32.0GB RAM and Ubuturn operating system.

### 4.4 Result Analysis

As shown in Table 2, we observe that our method outperforms both traditional machine learning methods and deep learning methods

on four different datasets. Compared to traditional machine learning methods, our approach uses deep neural networks to uncover nonlinear relationships of multi-view data, capturing the complex attributes of real-world data. As a result, we achieve significant improvements in all clustering metrics. Taking the Fashion dataset as an example, our model outperforms traditional methods with an increase of 19.9%, 22.6% and 20.1% for ACC, NMI and PUR, respectively. Compared to deep learning methods, our approach also demonstrates notable advantages, particularly on Reuters dataset, where ACC, NMI and PUR of our method exceed the best baseline by 6.2%, 7.7% and 12.1%, respectively. Furthermore, our method improves ACC and PUR by 5.1% and 5.8% on RGB-D dataset and achieves leading results on CCV and Fashion datasets.

To evaluate the effectiveness of our method for multi-view data with different number of views, we compare it with other MVC methods on Caltech-2V, Caltech-3V, Caltech-4V and Caltech-5V, respectively. As shown in Table 3, the clustering performances of most methods improve as the number of views increases, but EAMC and COMIC experience a performance decline. In contrast, our method exhibits significant improvements on all the four datasets, particularly on the Caltech-3V dataset, where ACC, NMI, and PUR increase by 5.8%, 3.8% and 6.7%. This is probably caused by that, our method comprehensively explores the local geometric structure between views, while achieving structure-level, instance-level and cluster-level consistency objectives simultaneously. These findings underscore the effectiveness and versatility of our multi-view clustering framework and highlight its ability for handling multi-view data from various domains.

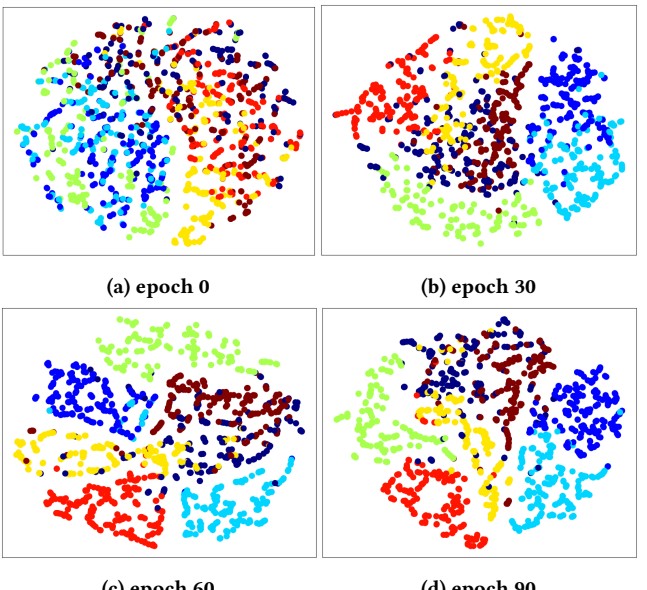

(a) epoch 0                     (b) epoch 30

(c) epoch 60                    (d) epoch 90

Figure 2: The $t-SNE$ visualization of the clustering results at different epochs on Caltech-5V dataset.

## 5 MODEL ANALYSIS

### 5.1 Visualization Analysis

In order to better visualize and analyze the performance of our approach, we employ the widely used t-SNE [6] tool to visualize clustering results on Caltech-5V dataset. We concatenate high-level features learned through instance-level contrastive learning and then map them to a two-dimensional space. In Figure 2, we can observe the scatter plots generated by our method at different iterations, with different colors representing distinct clusters. From these plots, it is evident to see that, as the number of iterations increases, the clustering structure becomes progressively clearer, further substantiating the effectiveness of our approach. The main reason is GC-CMVC achieves consistency at the graph level, cluster level and instance level, and obtains consistent clustering results across multiple spaces through self-supervised learning.

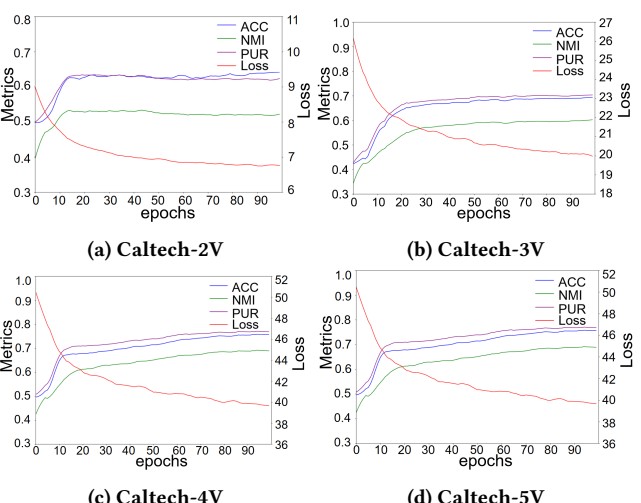

(a) Caltech-2V                  (b) Caltech-3V

(c) Caltech-4V                  (d) Caltech-5V

Figure 3: Loss and clustering evaluation metrics versus the variation of epochs on Caltech dataset.

### 5.2 Convergence Analysis

To demonstrate the convergence of our method, we conduct convergence analysis on four versions of Caltech dataset. We visualize the losses in the training process, along with the corresponding values of ACC, NMI and PUR. From Figure 3, we can observe that, the loss function rapidly decreases in the initial stages of training and then tends to converge to a stable value. Simultaneously, the values of the three metrics increase as the loss decreases, ultimately reaching stability. These results provide compelling evidence for the effectiveness and stability of our proposed model.

### 5.3 Parameter Sensitivity Analysis

In our objective function, there are two hyper-parameters, $\lambda_1$ and $\lambda_2$, which are used to balance the hierarchical contrastive learning loss and the graph learning loss. To gain a deeper understanding of how these two parameters affect the clustering performance, we conduct a parameter sensitivity experiment on Caltech datasets, and the values of ACC under different parameters are shown in Figure

| Datasets | Caltech-2V | | | Caltech-3V | | | Caltech-4V | | | Caltech-5V | | |
|---|---|---|---|---|---|---|---|---|---|---|---|---|
| Metrics | ACC | NMI | PUR | ACC | NMI | PUR | ACC | NMI | PUR | ACC | NMI | PUR |
| $L_{RC}$ | 43.7 | 31.6 | 42.7 | 47.6 | 32.4 | 47.6 | 55.1 | 46.6 | 55.7 | 63.2 | 53.7 | 62.6 |
| $L_{RC}+L_G$ | 59.1 | 34.9 | 57.3 | 54.9 | 36.7 | 53.6 | 69.4 | 56.2 | 67.2 | 74.9 | 68.2 | 76.1 |
| $L_{RC}+L_{HCL}$ | 56.4 | 42.5 | 58.1 | 62.5 | 43.3 | 62.8 | 70.9 | 63.1 | 71.1 | 80.1 | 73.4 | 79.3 |
| $L_{RC}+L_G+L_{HCL}$ | 63.2 | 50.5 | 62.5 | 68.3 | 58.7 | 69.1 | 75.7 | 70.8 | 77.4 | 83.2 | 75.7 | 84.6 |
| $L_{RC}+L_G+L_{HCL}+L_C$ | **64.2** | **53.5** | **64.1** | **68.9** | **60.4** | **70.6** | **76.4** | **72.8** | **78.4** | **84.9** | **76.4** | **84.9** |

**Table 4: Ablation study under different combinations of modules on Caltech dataset.**

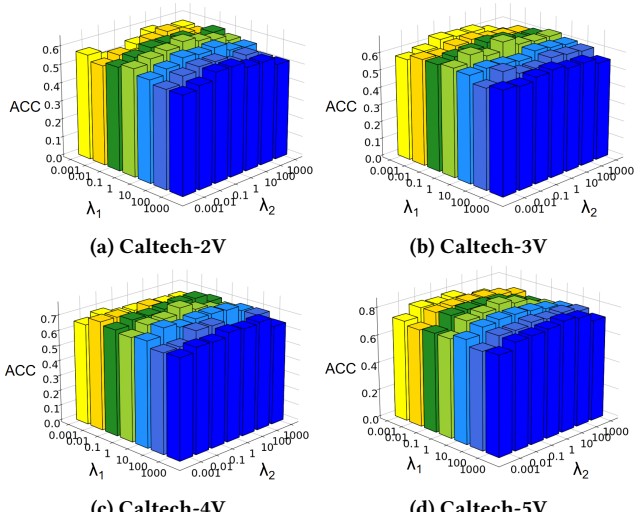

(a) Caltech-2V   (b) Caltech-3V

(c) Caltech-4V   (d) Caltech-5V

**Figure 4: ACC with different $\lambda_1$ and $\lambda_2$ on Caltech dataset.**

4. From Figure 4, it can be observed that, on Caltech dataset, our model achieves relatively stable ACC values under most parameter selections. However, we can also see that, too small or too large parameters will lead to slight decline of ACC. The best clustering results are all obtained with $\lambda_1 = 1.0$ and $\lambda_2 = 1.0$ on four versions of Caltech dataset, thus we set the parameters as this on all the datasets.

### 5.4 Ablation Study

To further validate the importance of four modules in our method, we conduct an ablation study on Caltech dataset. As shown in Table 4, combining the reconstruction module with the graph learning module or hierarchical contrastive learning module significantly improves the clustering performance compared to using the basic reconstruction module alone. The combination of hierarchical contrastive learning module and reconstruction module usually has better clustering performances than the combination of graph learning module and reconstruction module. It may be caused by that, hierarchical contrastive learning module can explore more types of consistency information. Additionally, integrating the self-supervised clustering module with other modules leads to improvements in various clustering metrics. The results of the ablation study indicate that, all the modules within the proposed method

play indispensable roles, and integrating these modules together can achieve the best performance.

## 6 CONCLUSIONS

In this paper, we propose a novel multi-view clustering framework based on graph consistency learning. Specifically, we achieve the preservation of feature and structure information through original feature reconstruction and intra-view contrastive learning. We perform consistency learning at the graph level, cluster level and instance level, simultaneously. Finally, we construct global features from high-level features to obtain a unified target distribution that guides all views towards clustering consistency by a self-supervised manner. It allows us to better capture the feature and structure information of multi-view data, leading to more accurate clustering results. Our method obtains outstanding performances across benchmark multi-view datasets, highlighting its wide applicability in multi-view data analysis.

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
