# OpenReview forum: "Graph based Consistency Learning for Contrastive Multi-View Clustering"
_acmmm.org/ACMMM/2024/Conference — MM2024 Poster_

### Official Review · Reviewer_sUQy · 2024-05-15

**Rating:** 5
**Confidence:** 3

**Summary:**

This paper proposes a novel multi-view clustering method based on graph consistency learning, where intra-view graph contrastive learning is used to uncover structural information within the view, and cross-view graph consistency is used to achieve structural consistency. A unified target distribution is also calculated to make all views have consistent clustering results.

**Strengths:**

1. Consistent graph learning is integrated into contrastive clustering to improve clustering performances, and an innovative clustering results matching method is developed.

2. The paper is well-organized and and easy to read.

3. Compared to state-of-the-art methods, the proposed method almost obtains the best clustering performances on a variety of datasets.

4. In the sections of parameter sensitivity analysis, ablation study and convergence analysis, the proposed method is fully discussed to make it better understood.

**Limitations:**

1. The comparison methods are a bit outdated, and the latest comparison method is from 2022.

2. There are some grammatical errors and typos in the paper.

**Suitability:**

3

---

### Official Review · Reviewer_2LMN · 2024-05-20

**Rating:** 5
**Confidence:** 3

**Summary:**

A novel multi-view clustering method termed GC-CMVC is proposed using graph consistency learning. In GC-CMVC, feature and structure information within the view is preserved by original feature reconstruction and intra-view contrastive learning, and the structural consistency among views is obtained by performing consistency learning at graph, cluster and instance levels.

**Strengths:**

1. This paper model the representation by intra-view and inter-view graph information is novel for existing contrastive multi-view clustering.
2. A novel clustering results matching strategy is developed. The global features are constructed at instance-level, and the clustering results obtained from global features and clustering-level contrastive learning are matched by a self-supervised learning method to guide consistent clustering results.
3. The proposed method is described detailedly and clearly. The paper is easy to read.
4. The experiment results demonstrate that the proposed method exhibits significant performance improvements compared to recent works.

**Limitations:**

1. This paper did not compare to other graph-based deep multi-view clustering methods.
2. The authors have obtained the probability distribution of cluster assignment Q. Why not consider utilizing it as the definitive clustering result? Does P (as defined in Equation 15) demonstrate superior performance compared to Q?
3. In the Ablation Study, the experimental results are not analyzed sufficiently to demonstrate the effectiveness of each module
4. Some parameters of neural network and its training are not provided, such as batch size and the dimension of the MLP.

**Suitability:**

3

---

### Official Review · Reviewer_mdYP · 2024-05-21

**Rating:** 5
**Confidence:** 3

**Summary:**

For multi-view clustering methods, to address the issue of neglecting the exploration of structural information within the view and the learning of structural information among views, this paper develops a new multi-view clustering method called GC-CMVC. GC-CMVC contains three modules: graph learning, hierarchical contrastive learning and self-supervised clustering. The graph learning module is integrated into contrastive learning framework to further explore the local information in each view and the consist information among different views. Experimental results on several multi-view datasets show that, compared with state-of-the-art methods, the proposed GC-CMVC obtains superior clustering performances. In the experiments, the convergence of GC-CMVC is demonstrated, and the optimal hyper-parameters are found. Through ablation experiments, the authors show that, all the modules of GC-CMVC are necessary, and removing any modules will lead to the decline of the performance.

**Strengths:**

1. To address the issue of partial information loss, this paper innovatively conducts contrastive learning at graph-level, instance-level and clustering-level simultaneously, focusing on both local and global information of multi-view data.
2. A novel clustering result matching method is developed to improve the clustering performance. The self-supervised clustering module is introduced into the framework of the proposed method, which is used to match the clustering results received from cluster-level contrastive learning and global features.
3. Different learning objectives are integrated into a unified framework, and they are balanced effectively.

**Limitations:**

1. The authors' proposed method did not have obvious advantages on RGB-D, CCV, and Fashion datasets.
2. Moreover, the training process of neural networks is not presented clearly.
3. In the experiments, the employed datasets are not very large. For example, the largest dataset only contains 18758 samples.
4. In Figure 2, for the T-SNE visualization of the clustering results of GC-CMVC, compared with 60 epchos, the clustering results of 90 epchos don’t seem to be any significant improvement.

**Suitability:**

3

---

### Official Review · Reviewer_BjWs · 2024-05-24

**Rating:** 5
**Confidence:** 4

**Summary:**

The paper introduces a novel deep multi-view clustering framework termed Graph-Based Consistency Learning for Contrastive Multi-View Clustering (GC-CMVC). This framework is designed to address several key limitations of existing deep multi-view clustering methods, particularly the neglect of structural information within individual views and the lack of structural consistency learning across different views. GC-CMVC integrates three main components: intra-view graph contrastive learning, cross-view graph consistency learning, and hierarchical contrastive learning. The intra-view graph contrastive learning module aims to uncover structural information within each view by creating relationship graphs and performing contrastive learning within these graphs. The cross-view graph consistency learning module encourages the alignment of structural information across different views, ensuring that the learned graph structures are consistent. The hierarchical contrastive learning module operates at both the instance and cluster levels, using separate feature spaces to capture fine-grained and coarse-grained information. Additionally, a self-supervised clustering module is introduced to enhance clustering consistency by generating global features from all views and employing self-supervised learning. The experimental results show that GC-CMVC outperforms several state-of-the-art methods.

**Strengths:**

1.The GC-CMVC framework introduces a unique combination of graph-based learning and contrastive learning to improve multi-view clustering. The integration of intra-view and cross-view consistency learning is a novel approach that addresses key limitations in existing methods.

2. The use of separate feature spaces for instance-level and cluster-level contrastive learning helps to capture both fine-grained and coarse-grained information, enhancing the overall clustering performance.

3. The authors conducted extensive experiments on various datasets, including RGB-D, CCV, Fashion, Reuters, and Caltech, demonstrating the superior performance of the proposed method compared to several state-of-the-art techniques.

**Limitations:**

1. In the final loss function, for four losses, only two hyper-parameters are employed, why not use hyper-parameters to balance L_{RC} and L_C?

2. For the proposed method, the construction of graph is important. However, in the paper, the value of the neighboring parameter for the graph is not discussed.

3. The comparison of run-time of different methods is not presented.

**Suitability:**

3

---

### Meta-Review · Area_Chair_YAGa · 2024-07-01

**Recommendation:** Accept (Poster)
**Confidence:** 5

**Metareview:**

This paper studies the structural information exploration issue in deep multi-view clustering. It uncovers intra-view structural information using graph contrastive learning and preserves the structural conscistency via cross-view graph learning. Further, it introduces two feature spaces to adjust the learning objectives, and utilizes  a self-supervision stragegy to promote the clustering consistency.
Experiments demonstrate its effectiveness.  After rebuttal and discussion, all reviewers recognize the contributions and are positive about this paper. So, acceptance is recommendated.